# Valorization of Local Legumes and Nuts as Key Components of the Mediterranean Diet

**DOI:** 10.3390/foods11233858

**Published:** 2022-11-29

**Authors:** Israel Hernández-López, Jordi Ortiz-Solà, Cristina Alamprese, Lillian Barros, Oren Shelef, Loai Basheer, Ana Rivera, Maribel Abadias, Ingrid Aguiló-Aguayo

**Affiliations:** 1IRTA, Postharvest Programme, Edifici Fruitcentre, Parc Científic i Tecnològic Agroalimentari de Lleida, Parc de Gardeny, 25003 Lleida, Catalonia, Spain; 2Department of Food, Environmental and Nutritional Sciences (DeFENS), Università degli Studi di Milano, 20133 Milan, Italy; 3Centro de Investigação de Montanha (CIMO), Instituto Politécnico de Bragança, Campus de Santa Apolónia, 5300-253 Bragança, Portugal; 4Laboratório Associado para a Sustentabilidade e Tecnologia em Regiões de Montanha (SusTEC), Instituto Politécnico de Bragança, Campus de Santa Apolónia, 5300-253 Bragança, Portugal; 5Department of Natural Resources, Institute of Plant Sciences, Agricultural Research Organization (ARO)—Volcani Institute, Rishon LeZion 7505101, Israel; 6Food Sciences Department, Faculty of Sciences and Technology, Tel Hai College, Upper Galilee 1220800, Israel; 7Miquel Agustí Foundation, Campus Baix Llobregat, 08860 Castelldefels, Spain; 8Department of Agri-Food Engineering and Biotechnology, Campus Baix Llobregat, Polytechnic University of Catalonia-BarcelonaTech, 08860 Castelldefels, Spain

**Keywords:** legumes, nuts, ready-to-eat, healthy, quality

## Abstract

Legumes and nuts are components of high importance in the diet of many countries, mainly those in the Mediterranean region. They are also very versatile and culturally diverse foods found all over the world, acting as a basic protein source in certain countries. Their genetic diversity is needed to sustain the food supply and security for humans and livestock, especially because of the current loss of habitats, species, and genetic diversity worldwide, but also because of the ever present need to feed the increasing human population. Even though both legumes and nuts are considered as high-protein food and environmentally friendly crops, developed countries have lower consumption rates when compared to Asia or Africa. With a view to increasing the consumption of legumes and nuts, the objective of this review is to present the advantages on the use of autochthonous varieties from different countries around the world, thus providing a boost to the local market in the area. The consumption of these varieties could be helped by their use in ready-to-eat foods (RTE), which are now on the rise thanks to today’s fast-paced lifestyles and the search for more nutritious and sustainable foods. The versatility of legumes and nuts covers a wide range of possibilities through their use in plant-based dairy analogues, providing alternative-protein and maximal amounts of nutrients and bioactive compounds, potential plant-based flours for bakery and pasta, and added-value traditional RTE meals. For this reason, information about legume and nut nutrition could possibly increase its acceptance with consumers.

## 1. Introduction—Legumes and Nuts as Mediterranean Staple Food

The conservation and use of legume and nut species is of increasing national and regional importance in the Mediterranean area. The genetic diversity of peas, chickpeas, lentils, fava beans, as well as almonds, hazelnuts, pistachios, and walnuts are needed to sustain food supply and security for humans and livestock. This is necessary because of the current loss of habitats, species, and genetic diversity worldwide, and particularly in the Mediterranean region, but also because of the ever present need to feed the increasing human population [1]. Feeding a constantly growing population with a demand for quality, health, and taste in food products is one of the great challenges of humanity. The manifestation of global changes, including climate, ecological, behavioural, and technological change, emphasizes the necessity to improve the current food production system to reduce the negative effect on the ecosystems [2]. An essential strategy to address food scarcity would be to use local varieties as a source of food health and environmental well-being. The use of native varieties as part of local food production can help create a more sustainable agricultural system [3]. The current food system focuses on researching more resources to increase the productivity of existing crops, while neglecting the study of crop diversity by incorporating more species. This results in the loss of agrobiodiversity, increasing susceptibility of the food industry to biotic and abiotic stresses, and the risk of catastrophic losses [4]. In fact, humans cultivate approximately 150 of the 30,000 edible plant species worldwide, with only 30 species being predominant in the human diet [4]. This explains the disparity between different populations, with 10% of people suffering from insufficient food consumption [5] and 2 billion people being overweight or obese [6]. That is why the use of native plants, legumes and nuts included, can help sustain food security. On the other hand, in industrialized societies, where a large part of the population has covered their minimum nutritional needs, the consumer asks for food not only for nutrients and taste, but also for a possible improvement in health or a reduction in the risk of disease. Consuming a varied, nutrient-rich diet is recognized as an essential part of a healthy lifestyle., Foods of plant origin have a particularly important role to play in healthy food consumption which is explained by their contribution of carbohydrates, fibre, minerals, vitamins, and a plethora of other non-nutrient components that appear to have a protective effect on many chronic diseases [7]. For this reason, Mediterranean varieties of legumes and nuts are reviewed here, with their reported nutritional benefits, and the potential to use them for novel food production.

For a better demonstration of quality, different European countries have created the terms Protected Geographical Indication (PGI) and Protected Designations of Origin (PDO) (Reg. EU No. 1151/2012). PGI identifies a product that originated in a particular place and where its relationship with the geographical area is essential for a given quality, reputation, or other characteristics of the product. At least one stage of production takes place in the PGI origin area. Products with a PGI quality scheme have specific quality characteristics that can be attributed to the production area. However, PDO identifies a product that originated in a specific place and its relationship with the geographical area is essential for the quality or characteristics of the product. All of the production phases of PDOs are carried out in that specific area. In addition, the environmental and human characteristics of that area (soil, climate, varieties used, traditional cultivation methods, etc.) have a direct influence on the properties of the product. Some examples of Mediterranean legumes and nuts which are classified as PGI or PDO products are Garbanzo de Escacena (Spain-PGI), Lenticchia di Castelluccio di Norcia (Italy-PGI), Lenteja de Tierra de Campos (Spain-PGI), Faba Asturiana (Spain-PGI), Bean “Haricot Tarbais” (France-PGI), Fasolina Kina Messosperma Kato Nevrokopion (Greece-PGI), Almendra de Mallorca (Spain-PGI), Castanha TerraFria (Portugal-PDO), Mongetal del Ganxet (Spain-PDO), Avellana de Reus (Spain-PDO), Hazelnut “Giresun Kalite” (Turkey-PDO) or Fagiolo Cannellino di Atina (Italy-PDO). Other examples with no PDO or PGI marks but from the Mediterranean area are: Blue Lupin (*Lupinus pilosus*) and Palestine Lupine (*Lupinus palaestinus*), which are Landraces species from Israel, Llargueta almond from Spain, Fagiolo di Controne from Italy, and “Osmanoglu” Chestnut from Turkey. Legumes and nuts are an essential component of the Mediterranean agro-ecosystem, from which one third of the “domesticated” foodstuffs for human consumption have been obtained [8]. Although many regions of the world are characterized by the Mediterranean climate and share a similar Mediterranean ecosystem, it is from the Mediterranean basin itself that the species used as grain for human consumption or fodder originates, and it is here that the cultivation of temperate legumes and nuts is born. The Leguminosae family is a major component of all habitats and sub-regions in the Mediterranean, with 91 genera, 1956 species, and 495 subspecific legumes. Over 1300 species are found in Turkey alone, of which about a quarter are endemic [9,10]. This makes the Mediterranean region a hotspot of diversity for legume species. On the other hand, nuts have been part of the human diet since prehistoric times and constitute an important element of food production in the countries bordering the Mediterranean Sea. In fact, the nut supply in Europe is highest in Mediterranean countries [11]. As there are currently local crops that are poorly exploited and marketed, either because they are considered “wild crops” or because they are harvested by hand by local farmers, promoting the use of local crops will not only improve environmental prospects, but could also increase avenues for trade and lead to the formulation of new, nutritionally enhanced foods, using alternative proteins and facilitating the intake of bioactive compounds and nutrients. This would make it easier to adopt them into Mediterranean diets through their use in plant-based analogues, bakery, and pasta flours and traditional ready-to-eat-meals with added nutritional value.

To cope with potential food shortages, modern agriculture and food systems focus on investing more resources to increase the productivity of existing crops, as opposed to increasing crop diversity by incorporating new species. In this review, ways to use local plants from different countries as food resources in current food products are explored, with a focus on ready-to-eat products. In addition to the rise in the consumption of healthy foods, the popularity of products recognized as beneficial is also on an upward trend. In addition, the growing concern for social and environment responsibility will also determine the future of these products. The aim is to formulate new convenience foods, promoting the dietary diversity and agricultural resilience of legumes and nuts, as well as the technologies available to increase their techno-functional range.

### 1.1. Legumes

Legumes are a fundamental part of the diet of millions of people due to their high nutritional value, being a fundamental pillar of the Mediterranean diet. Legumes have been part of the human diet since the earliest times of human society [12]. The most outstanding characteristic of legumes lies in their high protein content, both to produce animal feed [13] and for balanced human consumption [14]. Legumes are also a source of a large amount of minerals, such as calcium and iron [15], and are a source of energy. Ninety percent of the average calories consumed in developing countries are provided by plant-based products, among which legumes stand out, contributing nearly 10% of the total protein consumed by these populations [16]. Moreover, the role of legumes in improving soil fertility through the symbiotic association with atmospheric nitrogen-fixing bacteria are an essential key in sustainable agricultural rotation [17]. Broad beans (*Vicia faba*), chickpeas (*Cicer arietinum*) and lentils (*Lens culinaris*) are well adapted to the culinary culture of Asia and Europe [18]. Globally, by 2018, an average production volume of 92 million dried pulses was recorded, with chickpeas, lentils, and dried peas accounting for 70% of total production. The main producers were Canada, Russia, and the EU [19]. Broad beans, with 20% of the total world production, are among the legumes with the highest production, mainly in China, the EU, Ethiopia, and Australia. According to Lucas et al. [20], soybeans provide 70% of the vegetable protein supply in Europe.

The *Foods* journal included an introduction to the interest of legumes as a food ingredient in its special issue in 2020, also promoting their cultivation, characterization, and applications, since they don’t rely on an external nitrogen supply and therefore require less energy and have a lower environmental impact than other crops [21].

The Mediterranean region is the centre of diversity for the four major temperate legume crops: peas, chickpeas, lentils, and fava beans, which all originated in this region. However, cereal crop cultivation is prioritized by agricultural producers in Europe, which causes a decrease in production within the European Union. This decrease is due to the focus on intensive cereal and oilseed crops [22,23].

For example, in 2018, of all the European crop land, only 1.4% of it was used for legume crop farming, while cereals occupied a total of 31% of total land crops [24].

Wheat (*Triticum aestivum*) and other cereals predominate on European lands, whereas other types of crops, such as legumes, remain secondary [25]. Consequently, the European Union (EU) imports most of the plant proteins used for food production [26]. Moreover, modern consumption habits have radically decreased the consumption of pulses in Europe. This decline in legume consumption is directly related to the increase in the consumption of meat products in Europe, which went from 20 kg per capita per year in the 1960s to 50 kg per year in the 1980s, after which there has been a stabilization, hovering around 51 kg/person at present [27]. The increase in meat consumption meant a greater use of legumes for animal feed, but this was not reflected in their cultivation in Europe. Therefore, soybean imports were necessary to cover this demand in the industry. The latest Eurostat data are from 2018 and account for 4.11 million tons in the EU-27 (excluding production destined for the UK). In its latest agricultural market outlook report, the European Commission called for an increase in protein and legume production in the EU correlated with higher demand for human consumption, mainly due to changes in the consumption habits and in animal feed [28]. However, in countries such as Canada and Australia, the production of pulses has had a marked increase, this possibly being the main reason for the production decline of pulses within the European Union [27].

### 1.2. Nuts

Nuts are commonly consumed in the Mediterranean diet, but their intake is popular worldwide [29]. The most popular and commercialized are peanuts (*Arachis hypogaea*—interestingly, a legume), almonds (*Prunus dulcis*), cashews (*Anacardium occidentale*), Brazil nuts (*Bertholetia excelssa*), hazelnuts (*Corylus avellana*), macadamia nuts (*Macadamia integrifolia*), pecan nuts (*Carya illinoinensis*), pine nuts *(Pinus pinea*), pistachio nuts (*Pistachia vera*), and walnuts (*Juglans regia*) [30].

The production of edible nuts and dried fruits in Europe has augmented since 2010, while in the same period imports have had an annual growth rate of 3%, representing 2.9 million tonnes by 2017 [11]. The most-produced edible nuts in Europe are hazelnuts and almonds. In terms of primary production (growing, harvesting, and drying), the largest European producers are Spain (with almonds leading their nut production), and Italy (hazelnuts) [11]. At a global level, Spain is the third-largest producer of almonds in the world. Italy is second in hazelnut production. France is the seventh-largest producer of walnuts. The Netherlands and the United Kingdom are large processors of imported nuts (the main one being peanuts). Imported nuts are widely used by the confectionery, baking, and sweets industries for further processing (roasting, mixing, coating, and frying). Motivated by a constant tendency towards healthier lifestyles, it is expected that European consumers will increasingly appreciate edible nuts in the future. Although Europe produces only 10% of the world’s production of tree nuts, it was the largest consumer, with 25% of the total world consumption in 2020 [11].

## 2. Classification, Nutritional and Technological Functionality of Legumes and Nuts

### 2.1. Legume Characterization as a Healthy Food Source

Legumes are particularly rich in carbohydrates and proteins, comprising one of the large groups of organic compounds found in nature. For this reason, legumes play an important role in the human diet, providing the protein, energy, dietary fibre, minerals, and vitamins required for human health [31]. Moreover, they provide an important source of essential amino acids when consumed with cereals and other foods rich in sulphur-containing amino acids and tryptophan [32]. Moreover, legumes contain a high amount of phytochemicals [31], biologically active substances that are not essential for life but have beneficial regulatory effects on health, e.g., protection against cancer, cardiovascular protection, maintenance of gastrointestinal health, etc. This justifies their increased consumption in diets in special situations and in the diets of people with various pathologies. On the other hand, some anti-nutritional compounds were found in some legumes, such as alkaloids in lupins, or lectins in kidney beans; these may be bitter, allergenic, or even toxic.

Anti-nutrients are natural or synthetic compounds that interfere with the absorption and digestion of nutrients. Substances derived from secondary metabolism such as saponins, phytic acid, protease inhibitors (such as trypsin), tannins and oxalates have also been found. These compounds could exert detrimental effects on the health of consumers by interfering with digestion processes, the activity of enzymes, bioavailability, and intestinal absorption of nutrients [32]. However, the elimination of the anti-nutritional compounds can be achieved by the selection of plant genotypes or through post-harvest processing techniques [33] that are constantly improved [34].

Common beans (*Phaseolus vulgaris* L.), thanks to their high protein (17.96–23.62%) and carbohydrate (56.53–61.56%) content, are the most consumed beans worldwide. Moreover, despite their low-fat content (1.27–3.62%), they have been reported to contain essential fatty acids such as linoleic (C18:2, cis-∆^9,12^) and linolenic (C18:3, cis-∆^9,12,15^), the precursors of *n*-6 long chain polyunsaturated and *n*-3 long chain polyunsaturated fatty acids (PUFA) [35]. They also present a balanced amino acid composition. Beans are an excellent source of minerals. They also have a high folate content and a balanced content of other B vitamins except for B12. On the other hand, their consumption is negatively affected by reduced protein digestibility and anti-nutritional factors, such as proteolytic enzyme inhibitors, alkaloids, phytic acid, and lectins [33].

Soybean is widely used in the food and feed industry because of its high protein and oil content [36]. Soybean is the primary source of vegetable oil and protein products, such as defatted soy flour and soy protein concentrate. Edible soybean is either consumed directly or processed into various products which can also be derived into plant-based meat alternatives. The composition of soybean may vary somewhat according to its origin and growing conditions. Through plant breeding, it has been possible to obtain high protein (40–45%) and lipid (18–20%) levels [37]. Usually, an increase of 1.0% in protein content is correlated to a decrease of 0.5% in oil level. The four major fractions of soybean proteins are known as 2S, 7S, 11S, and 15S. The 11S and 7S fractions constitute about 70% of the total protein in soybean [37].

Their consumption is also negatively affected by reduced protein digestibility and anti-nutritional factors. However, legumes such as soybeans have also been shown to help prevent certain types of cancer not only because of the presence of antioxidants, but also due to the content of certain protease inhibitors, which can inhibit the enzymes MMP-2 and MMP-9, thus potentially reducing the progression of cancer [33]. On the other hand, the application of fermentation of legumes brings great advances such as the reduction of their anti-nutritional factors, promoting nutritional digestibility and reducing their allergenicity. Soya is the most widely used legume for fermentation, but there are also other unexplored legumes with great potential for the production of fermented foods that contribute to the prevention of allergenicity. Fermented foods also contribute to the prevention of cardiovascular diseases.

Peas contain five species depending on the taxonomic interpretation [38]. *Pisum sativum* (the field or garden pea) is domesticated and is a major human food crop worldwide. Like other legume seeds, pea is rich in protein (18–30% of dry weight) and contains vitamins, minerals, and dietary fibre. In fact, an average serving of peas has almost the same protein content as a whole egg (with lower quality), less than one gram of fat, and no cholesterol. They also have a significant mineral content, such as phosphorus, iron, potassium, and magnesium. In terms of vitamins, it contains thiamine, niacin, folates, and mainly vitamin C (although a considerable part of it may be lost during the cooking process) [38]. The storage proteins of pea grains are composed mainly of legumin (11S), vicillin (7S), and albumin (2S), and most pea protein isolates contain globular 11S and 7S. The ratio of legumin to vicilin in peas ranges from 0.2 to 1.5 [38].

The average total protein content of white lupins (*Lupinus albus*) varies between 30% and 42%, depending on the lupin variety. Approximately 87% of total lupin protein is stored in globulins, also known as conglutins. When compared to other seeds or legumes, lupin has protein levels ranging from 30–42%, reaching levels like those of soybeans (38–42%), indicating a suitable option to add to food, promoting an increase in protein and dietary fibre levels. It also has physical properties like those of soybeans, such as appearance and size [34]. Among legumes, lupin is distinguished by its high dietary fibre content, ranging from 37.5 to 40.2% on a dry basis, which is unique compared to other legumes, including soybeans [39]. The oil content of lupin may range from 1.0 to 17.0%, with a high variation in fatty acid composition. Their high carbohydrate and protein content makes them a perfect source of energy for those who engage in physical activity. Lupin seeds, like other legumes, are important sources of vitamins and phenolic compounds. For example, carotenoids and tocopherols are present, with the former being mainly responsible for the colour of the oil fraction.

Legume seeds represent an essential portion of the human diet not only for their protein source, but also for their bioactive compound contribution [40,41,42]. Legumes are an important source of bioactive phenolic compounds which play a significant role in many physiological and metabolic routes for the human health and are important determinants of the color, taste, and flavor of foods. Bioactive phenolic compounds exhibit free radical-scavenging capacity and the ability to interact with other compounds such as proteins. The bioactive phenolic compounds present in grain legumes (as reactive metabolites and associated antioxidant activity) make them suitable candidates for creating new functional foods [14]. The principal phenolic compounds that are present in legume grains are phenolic acids, flavonoids (mainly catechins and procyanidins), and condensed tannins, present principally in the legume seed coats. Kidney bean and mung bean present principally gallic and protocatechuic acids. Catechins and procyanidins characterise ca. 70.0% of total phenolic compounds in lentils (seed coat) [43]. The antioxidant effect of phenolic compounds is directly correlated with their chemical structures such as the number and position of the hydroxyl groups. The cooking and processing of legumes leads to the reduction of phenolic compounds owing to chemical rearrangements. The health benefits of phenolic compounds include acting as an anticarcinogenic, antithrombotic, antiulcer, antiatherogenic, antiallergenic, anti-inflammatory, antioxidant, immunomodulating, anti-microbial, cardioprotective, and analgesic agents [43].

A growing interest has been recently shown for technological functionalities of legume proteins in view of their use as food ingredients, mainly as alternative substitutes of dairy, meat, egg, and wheat proteins. However, legume proteins usually lack some important properties (e.g., emulsifying, gelling, whipping, water absorption properties, and oil binding capacity) required to provide food matrices with good quality properties, especially in terms of structure. Furthermore, the specific physicochemical characteristics of each protein (e.g., hydrophobicity and structural flexibility), also environmental factors (e.g., pH, ionic strength, and temperature), solvent-type, and treatments (e.g., during-extraction or post-extraction) can affect techno-functionality, mainly modifying dispersibility and solubility. Indeed, well dispersed or solubilized proteins usually show better technological properties [44,45].

Chickpea, fava bean, lentil, pea, lupin, and soy protein isolates produced by different methods (i.e., isoelectric precipitation, salt extraction, alkaline extraction followed by acid precipitation) showed good emulsifying properties, due to their ability to be adsorbed at the surface of oil droplets, thus reducing the interfacial tension and preventing coalescence [44,46,47]. Zhao et al. [45] demonstrated that a commercial pea protein concentrate had emulsifying, foaming, water absorption, and pasting properties similar to a commercial soybean isolate. The foaming capacity of Australian sweet lupin isolates prepared by alkaline extraction at pH 9.0 followed by acidic precipitation at different pH levels showed higher foaming capacity and foam stability than a soy protein isolate, especially under acidic conditions [47].

The legume species can affect the technological properties of the proteins. For instance, better gelation properties were found for white lupin (*Lupinus albus*) compared to blue lupin (*Lupinus angustifolius*) protein isolate [46]. The genotype also proved to be relevant for bean technological properties and behaviour during extrusion [48].

To improve the technological functionalities of legume proteins, different physical and (bio)chemical treatments can be applied. Heat treatment is one of the most frequently used, being able to change protein and starch organization. Peng et al. [49] obtained pea protein samples with different degrees of aggregation by heat treatments at different protein concentrations and tested the emulsifying properties. The authors demonstrated that heat treatment resulted in inter-droplet hydrophobic interactions in emulsions, increasing the droplet flocculation and creating a gel-like network that increased emulsion viscosity and creaming stability. Extrusion-cooking was successfully applied to bean powders for the design of specific technological properties [48]. Enzymatic hydrolysis can be an alternative to heat treatments. Nawaz et al. [50] showed that a partial enzymatic hydrolysis of a fava bean protein concentrate increased the colloidal stability of UHT-treated oil-in-water emulsions, reducing droplet size, polydispersity, coalescence, and flocculation indexes. The effectiveness of the enzymatic treatment was related to the size of the obtained peptides, showing a more flexible structure, and higher surface charge and hydrophobicity. The degree of hydrolysis and the enzyme used for the treatment proved to be important factors in changing pea protein techno-functionality [51]. Fermentation is another key process that can enhance the technological properties of legumes, while favouring the elimination of off-flavours and the improvement of nutritional features, as reported in the review by Garrido-Galand et al. [52]. Other valuable alternative technologies can be ultrasound, high pressure, gamma-irradiation, microwave, and pulsed light [53], but more research efforts are needed to elucidate the factors affecting legume protein techno-functionalities and the effects of the different modifying technologies.

### 2.2. Nuts Characterization

Edible nuts are cultivated in a wide range of growing conditions and climates. They are especially valued for their sensory, nutritional, and health attributes, contributing to energy and nutrient intake directly (e.g., tocopherols, magnesium, potassium) and indirectly via multiple mechanisms [54]. Each form of nut has its own characteristic sensory profile, which is reasonably attractive to individual consumers and so will influence their acceptance [55]. However, the sensory profile of raw nuts is commonly modified through processing, such as roasting and frying, which blackens the colour, increases brittleness, and develops new flavour compounds. A wide selection of flavorings (e.g., salt, sugar, cinnamon, and capsaicin) are also commonly added directly to the surface of nuts to enhance their appeal [56,57]. Broadly, such modifications increase sensory variety and, and by having a more appetizing appearance, may facilitate regular nut consumption and intake of the nutrients they contain [55]. The sensory, nutrient, and/or physical properties of nuts alter gut hormone secretion and appetitive responses by consumers [58]. In addition, nuts are commonly integrated into the matrix of other foods (e.g., confectionery, baked goods, ice cream), altering the flavour profile and producing a single new combined sensory incentive that may guide intake of that item or influence the acceptability and selection of other items in the wider diet [55].

The main component of nuts is their quality fat (53–60%), which makes them concentrated sources of energy: 100 g of edible part of nuts provide 500–600 kcal. Moreover, nut seeds are notable for their high content in monounsaturated (MUFA) and polyunsaturated (PUFA) fatty acids (42.88–66.71%), being > 75% of the total lipids, protein (7.50–21.56%), tannins (0.01–0.88%), phytate (0.15–0.35%), vitamins E and K, folate, thiamine, minerals (such as magnesium, copper, potassium, and selenium), and substances such as xanthophyll carotenoids, antioxidants, and phytosterols compounds, with recognized benefits to human health. In fact, nut seeds with skins can be a good source of fibre (0.55–3.96%) [59]. Edible nuts present a low moisture content (1.47–9.51%), making them a stable food product. With respect to fatty acid composition, oleic acid (C18:1) was shown to be the main component within monounsaturated lipids among the samples, while linoleic acid (C18:2) was de major constituent for polyunsaturated lipids, except for macadamia nuts. In addition, linolenic acid (C18:3) was shown to contribute as the second major constituent of polyunsaturated fatty acids, ranging from 11–13% and linoleic acid up to 59.8%, which help to control triglyceride and cholesterol levels in the blood [59]. Walnuts contain omega-3 fatty acids, precursors of DHA and EPA [59]. Various analyses of amino acid composition indicated that lysine (Brazil nut, cashew nut, hazelnut, pine nut and walnut), methionine, cysteine (almond) and tryptophan (macadamia, pecan nut) are the amino acids that are least consumed in a regular diet among people from 2 to 5 years of age, so the consumption of legumes and cereals is suggested to obtain the required adequate protein intake. The amino acid composition of the seeds was characterized by the dominance of hydrophobic (37.16–44.54%) and acidic (27.95–33.17%) amino acids followed by basic (16.16–21.17%) and hydrophilic (8.48–11.74%) amino acids. Trypsin inhibitory activity, hemagglutinating activity, and proteolytic activity were not documented in the scientific literature [59]. In this context, prior reviews and epidemiological and/or clinical trials, such as the PREDIMED study concerning high-fat diets based on olive oil and nuts, suggested that the regular consumption of these products has several health benefits in terms of obesity, hypertension, diabetes mellitus, and cardiovascular diseases, with a reduction in mediators of chronic diseases such as oxidative stress, inflammation, visceral adiposity, hyperglycaemia, insulin resistance, endothelial dysfunction, and metabolic syndrome [7,60].

Technological properties of nuts have not been deeply explored so far. However, the active use of nut flours in food formulations requires a better knowledge of emulsifying, gelling, whipping, oil/water holding/binding capacity, etc. For instance, a high oil holding capacity is preferable for stabilizing food products with high fat content, including emulsions; a high water-holding capacity can prevent syneresis and provide a thickening effect in some food products [61].

The low amount of starch completely changes the pasting properties of nut flours with respect to wheat flour, with relevant effects on bakery product texture. Similarly, the lack of gluten-forming proteins is an issue in the development of baked goods or pasta. Thus, a deeper knowledge about the techno-functionality of nut flours is fundamental to improving their use, and especially to valorize nut press cakes, which are the remnant after the extraction of oil from the nuts, deriving from the oil industry [62,63].

## 3. Ready-to-Eat Products

Ready-to-eat food (RTE) is that which doesn’t need any further ingredients or preparations to be eaten. In most cases it just takes a minimal amount of heating to be ready. Today, these products have gained great interest among consumers because of their convenience and attractiveness, and above all, because of the time they save due to their stability, even in long storage periods [64]. The RTE definition covers both non-prepacked and prepacked products, and it is intended to apply whether the RTE food may be consumed hot or cold. The expression “without undergoing any further treatment” does not include dish preparation activities such as light washing, slicing, chopping, portioning, marinating or preservation, on a food product ready for consumption, but to emphasize that it does not hav eto be cooked in order to be eaten [64]. Under this definition, several processed foods can be regarded as RTE products, including snacks (e.g., biscuits or crisps), breads, pies, sandwiches, dairy products (beverages, milk, cheese, spreads), prepared salads, and minimally processed vegetables and fruits. The list can be extremely long, and new products enter the food market nearly every day. This wide range of possibilities makes it possible to target the product to all possible audiences, or to screen it for specific groups of people such as the elderly or babies.

On the other hand, high consumer demand in the West for healthy and vegetarian products has become an important global concern [42]. Epidemiological studies suggest that regular or increased consumption of fresh vegetables may reduce the risk of chronic diseases [65]. Because of the growing consumer demand for healthy, natural, and convenient foods, attempts are being made to improve RTE foods’ nutritional values via modifying their nutritive composition. In some cases, the consumption of RTE products could be considered as processed food, and abundant literature evidence suggests that ultra-processed foods often affect human health negatively [66]. Even though, according to Fardet, [67], the less the food is processed, the higher its satiety index is and the lower the glycemic index becomes, therefore, minimally processed RTE foods could represent a healthy option for consumers. In this context, the application of legumes and nuts, with a high content of protein and lipids, can be a feasible option to improve the formulation of RTE products as healthy processed foods. In fact, the food processing industry is developing newer uses of these seeds and nuts for producing food products having beneficial effects on human health [14]. Table 1 shows the major RTE products mentioned in the scientific literature in recent years, where legumes and/or nuts are the focal ingredient. The formulation, processing, and target audience are also reported.

### 3.1. Methodology

A systematic literature review was conducted to search for ready-to-eat (RTE) products formulated with legumes and nuts. For the writing of Section 3 of the present manuscript, the PRISMA (Preferred Reporting Items for Systematic Reviews) and meta-analysis method was used to select works on the formulation of RTE products based on legumes and nuts from scientific databases with clear inclusion and exclusion criteria and quality assessment. A bibliometric study was carried out on the pre-selected papers for meta-analysis in terms of geographical distribution, affiliation, and main citations in SciELO, Scopus and Web of Science (data not shown). Experts in the field were contacted for information on any other relevant interventions not identified by the electronic search. The keyword search was “Ready-to-Eat” AND formulation AND legume AND/OR nut, with their respective combinations. The searches were performed with the title, keywords and abstract of the papers in different databases separately. The searches identified studies published from January 2005 to October 2021. Only peer-reviewed articles, and articles written in English, were considered. Only papers focusing on the formulation of RTE products using legumes and nuts were included. All articles referring only to the nutritional analysis of RTE products where neither the formulation nor the description of the production process was specified were excluded. The databases provided the results from 461 publications with the keyword search, from which duplicates were removed. These duplicates consisted of the same articles with varied reference formatting of author details. Once the duplicates were removed, the search results were examined for articles related to the specific topic and related only to the nutritional, physicochemical or sensory evaluation of the topic. Thus, from the total of 461 articles, 216 articles were excluded that did not belong to the area of interest and whose full articles were not available. Attention was focused on 21 articles found in the screening process. From a detailed study of these articles, the formulation of different RTE products targeting different consumers and for different meals of the day was found.

### 3.2. Enriched Food

The wide range of possibilities in the definition of RTE products also includes a broad range of the target audience. Scientific articles have been found with new products formulated for infants, malnourished children or elderly people, enriched with ingredients derived from legumes and nuts. The term ‘plant-based’ is used to describe a recent consumer trend of avoiding animal-based products and choosing healthier plant-derived alternatives instead. For example, Bassey et al. [68] formulated a RTE weaning food with a good protein (16.89%) and oil (8.38%) content, including octadecenoic and octadecadienoic acid, and making it a good supplementary food that allows babies to move from liquid to solid foods. For the elderly, RTE products such as drinks, instant soups, or snacks have been formulated with soybean and mung bean flour [72]. These products were balanced in energy distribution from macronutrients and contained good quality protein, as well as being low in saturated fatty acids and free sugar.

### 3.3. Snacks

Several scientific papers were found related to snack processing, including nut and legume-based products [78,79,80,82,83,84,85]. In fact, population-based studies have shown increased food consumption related to the snacking habit. Breakfast, afternoon, or aperitif snacks are popular and convenient foods and, therefore, they would be ideal food formats to deliver plant-based products. Among the snack foods, extruded snacks have the greatest growth potential. Extrusion technology has led to more diverse and complex formulations for snack foods compared to other processing methods and it has been used in many industries to produce new and unique snacks. Extrusion is a high-temperature short-time (HTST) process which involves simultaneous thermal and pressure treatment along with mechanical shearing, resulting in changes such as gelatinization of starch and the denaturation of proteins. These changes increase the digestibility of both carbohydrates and proteins, making the extrusion process a viable and advantageous option for RTE-foods production [89]. Many extruded products are mostly made from cereals such as corn, rice, and wheat, but they can include other ingredients, such as legumes and nuts, leading to several nutritional benefits such as high protein and fibre content, low-calorie content, and their gluten-free nature. The extrusion process has also been shown to reduce anti-nutrients presents in legumes, such as trypsin inhibitors, lectins, and phytic acids. [89]. In fact, some extruded snacks with legumes [81] and nuts [71] can be found in the literature. For example, Alam et al. [78] used lentil flour in the formulation of a RTE nutritious snack mix for “on-the-go” consumption. Other legumes, such as black gram [81] and common beans [84] were also used, with the aim to elaborate a nutritive bar snack with good acceptability.

### 3.4. Meat Analogues and Their Derivatives

Given the trend to look for healthier food alternatives, legumes, and nuts, thanks to their main nutrients, shows a high potential for applicability in RTE products, especially to substitute foods containing gluten or meat and their derivatives. Among these substitutes, there are cheese-like products obtained from soy (such as tofu) or nuts. In particular, “vegan cheeses” are cheese analogues made from different types of nuts (i.e., cashews, macadamias, almonds, etc.) obtained after soaking, grinding, and water addition. The homogenate is completed with other ingredients (e.g., spices, herbs, lemon juice, salt, etc.) depending on the recipes and regions, and undergoes coagulation with different coagulant solutions or fermentation processes. These changes lead to achieve a final product with unique sensory and qualitative characteristics. In this context, Oyeyinka et al. [73] formulated cheese analogues with different concentrations of soy and cashew nuts, reporting that the overall acceptability of the samples decreased with increasing cashew nut concentrations. Specifically, the substitution of soy drink with 40% cashew nut drink for the vegan cheese analogue production resulted in a product with desirable acceptability [73]. On the other hand, the use of legume proteins has also been studied with the objective of formulating egg-free RTE vegan products. Indeed, native and modified proteins deriving from peas, lentils, lupines, and chickpeas can confer interesting technological functionalities, such as gelling, emulsifying, and foaming properties. Aquafaba, the water derived from legume cooking, is another ingredient gaining interest as an egg substitute for bakery products or mayonnaise due to its foaming, emulsifying, thickening, and gelling properties [90,91]. As an example, Raikos et al. [92] observed that chickpea aquafaba exhibits emulsifying properties due to its composition, namely protein, water-soluble/insoluble carbohydrates, coacervates, saponins, and phenolic compounds. The addition of hydrocolloids was also suggested to improve the technological functionality of proteins intended to formulate egg-free products, as it improves protein solubility.

Legumes are also an interesting source for the extraction of plant proteins due to the high protein content. It is essential to select the appropriate processing for plant proteins extraction to maximize the yield and to enhance the applicability in the food industry. Pea and soy protein isolates are the legume proteins mostly used in RTE products because of their low processing cost, functionality, and availability, especially in meat analogues due to their high albumin and globulin content that allows for the creation of solid building blocks [93]. However, the use of mung bean (*Vigna radiata*) is currently gaining popularity for the elaboration of RTE foods. In 2021, the protein isolate of mung bean (MPI) has been approved in the novel food category (Regulation (EU) 2015/2283). Although mung bean and mung bean flour have a long history of food use in Asia, there is no history of use for the protein isolate. Mung bean was used in the literature to produce some RTE products with a good overall quality acceptance [72,79]. Prasad et al. [86] elaborated a mix with cereal-mung bean fulfilling one third of the recommended protein and energy requirements of children under the age of five years.

All plant proteins could be good candidates for the preparation of RTE products, especially meat analogues and their derivatives. To this aim, disruptive processes are necessary for the transformation of native globular proteins into unfolded and filamentous aggregates, transforming them into even more technologically functional components in terms of solubility, gelation, emulsification, and foaming capacity. Diverse processes have been applied in order to obtain both improved nutrimental intake from legumes and anti-nutrient free products. As most of the anti-nutrients present in legumes can be degraded by heat application, water-boiling or roasting legumes could be a suitable option to reduce processing times and enhancing nutritional intake [94,95,96,97,98]. In this respect, soybean and pea are commonly used due to their high albumin and globulin content. The stored globulin, accounting for as much as 90% of total proteins, is divided into two fractions according to their sedimentation coefficients (the 7S and the 11S fraction). The β-conglycinin and vicilin are the main components of the 7S fraction and glycine and leguminin are the main components of the 11S fraction in soy and pea, respectively. The gel-forming ability of some proteins present in meat can provide a particle binding role, immobilising fats, and trapping a high-water content within the matrix of emulsion-type alternative protein products. Table 2 shows the main technologies used on different types of legumes to change the technological functionality of vegetable proteins and achieve new properties for the elaboration of RTE foods, especially meat analogues.

### 3.5. Beverages and Soups

The wide range of RTE products also includes some beverages derived from aqueous extracts of plant ingredients, such as plant-based milk-like products. According to the scientific literature, in the US, an increase in digestion problems of lactose has appeared in almost 65% of the population over the last decade. In this context, plant-based alternatives are gaining popularity, with an increased consumption of 60–70% [113]. So far, vegetable substitutes of milk have been based mainly on soybean. However, the range of possibilities is growing exponentially, including the use of legumes and nuts. Plant-based milk alternatives are prepared by disintegration of plant material, which affects the particle size and the stability of the final product, depending on the nature of the raw material, the method used for disintegration, and the storage conditions [114]. Despite of the popularity gained by plant-based beverages, bioaccessibility is still an issue, as cow milk nutrimental intake is not covered and nutrients such as calcium, vitamins and minerals can be restricted by the presence of anti-nutrients such as phytates and oxalates. However, fermentation by lactic acid bacteria (probiotics) could enhance calcium and vitamin intake from plant-based milk substitutes produced with legumes and nuts [113].

Other liquid or semi-liquid RTE foods are soups, with flavourful and high nutritious composition and that are usually served at the beginning of a meal or as a snack. When the main ingredients come from legumes, they usually have a predominant flavor, and sometimes other seasonings are used to mask this flavor so that it is not unpleasant. Soups are broadly classified into different types based on the texture (i.e., thick, and thin or clear soups). Thick soups are classified depending upon the type of thickening agent. For example, purées are vegetable soups thickened by starch. Alvarez et al. [77] developed a new “ready-to-eat” semi-solid chickpea soup by using high hydrostatic pressure (HHP) at 600 MPa and 50 °C for 15 or 25 min combined with final microwave heating prior to consumption. HPP induced starch gelatinization, leading to the characteristic rheology and texture of purée or cream. The development of this refrigerated gluten-free HHP-induced legume-based product, which can be given a quick final heating in a microwave oven, would be commercially interesting and foreseeably successful, providing the catering industry and consumers with various new soup/cream products with high protein content (19.4% of the total product).

## 4. Market Projections and Future Perspectives

The changes in food consumption trends are going to have a significant and direct influence on food production design in the long and short run [115,116,117]. This situation generates unique opportunities to marketers and governments to provide sustainable food products to populations regardless of their geographic regions and differences in household incomes. With the changing lifestyles of consumers, with rising employment opportunities and increased tourism, the demand for RTE food products is projected to grow soon [118,119]. Many studies affirm that RTE foods, which have proved to be a competitive source of nutrients, are very popular and are consumed greatly by youth [120,121]. With the global population projected to increase above nine billion by 2050, we have come across a great opportunity to produce and distribute adequate healthy food to all of humankind [122]. In this context, marketers should anticipate different approaches to sustain consumer interest in their products offered in the market, combining convenience with healthiness. The increasing concern of consumers for the welfare of animals, the environment (e.g., climate change, carbon footprint, the limitation of the planet’s resources, etc.) as well as the tendency to take care of their health and their emotional-mental well-being are the main drivers of change in the food system. It has been noted that plant-based RTE products are a low-power product category, but can be adapted to current and future needs [123]. As an example, in the US, the amount of plant grains fed to livestock is enough to feed more than 800 million people who follow a mainly plant-based diet [124]. Therefore, the distribution and use of food needs critical rethinking and transformation.

The current trend towards the consumption of plant-based proteins is at the center of several objectives in terms of environmental care and the fight against malnutrition at a global level. In fact, proteins are named for ensuring nutrition and health for people from underdeveloped countries. As mentioned above, animal-based products and animal proteins are the subject of focus for their high environmental and climate impact, and the eutrophication of water due to intensive agricultural production, mainly serving livestock farming [58]. Generally, meat production’s effects on greenhouse gas emissions is typically larger than even the most inefficient production of fruits and vegetables [125]. On the other hand, legume and cereal-based protein products offer a valuable choice for both vegetarians and meat consumers. However, plant proteins occasionally lack the presence of certain amino acids. Whole grains are generally low in lysine content and high in methionine. Conversely, legumes have a high lysine content, while methionine levels are low [126]. Therefore, a dietary pattern or RTE products including both legumes and grains can be beneficial in providing enough of the essential amino acids. For that reason, proteins from soybean, peas (legumes), and wheat (cereal), commonly used to create solid building blocks due to their low processing cost, functionality, and availability, are often combined with rice proteins [93]. Soy proteins remain by far the most widely used plant-based ingredient in foods, but demand is also high for newer ingredients. According to *Euromonitor*, the most promising future protein sources are peas, hemp, seaweed, and ancient grains, such as chia, buckwheat, and amaranth. It singles out peas and hemp as compelling alternatives to cow’s milk and dairy ice cream, while other plant proteins may be better suited as secondary ingredients in savoury snacks and baked goods. Meanwhile, Leatherhead Food Research says soy accounts for 56% of the global market, but pea proteins are among the fastest growing plant proteins in new product launches.

Companies are exploring cost-effective sources of plant proteins and ways to increase protein extraction yields by modifying extraction processes and technologies, with the objective of creating novel food products. In the past, only cereal-grains and soybean were used for the development of extruded food products. In recent years, several studies have considered the incorporation of other legumes instead (such as bean, lentil, pea, chickpea, and fava bean) to improve the nutritional value of extruded snacks [127]. Over the past 10 years, numerous articles have been published containing legume extrusion as the main topic, showing an excellent potential to produce extruded RTE foods [89].

In this context, continued growth and demand in the rapidly and healthy RTE market will provide a wide range of opportunities for food scientists and entrepreneurs to explore novel innovations. Texture-related sensory properties and the disguising of the pronounced taste of some legumes are the primary targets for product development and applied research [128]. However, these problems have not been satisfactorily resolved in the use of plant proteins, so more rigorous studies and food applications are needed.

However, we must not leave aside the fact that, although as it has already been demonstrated that legumes and nuts are a great source of nutrients and are of great versatility, dietary changes can always present some obstacle to their consumption. It will always be necessary to evaluate the digestibility of the nutrients present, as well as to consider that this can generate new allergic processes and trigger cross-reactions. This is of particular importance considering the consumer trend towards to a more plant-based diet.

Therefore, there are challenges for the market to sustain its current momentum due to above-mentioned technological barriers as well as factors regarding product safety, nutrition, shelf-life, and regulations [129]. A systematic scientific approach to investigate the technological functionality of legume and nut nutrients, physicochemical interactions between various ingredients, and the influence of processing conditions is needed to continue to address these challenges. As a final note, plant-based protein products are a dietary option; it is unlikely that they will replace regular meat and poultry products. For this, the right approach for scientists and food processors is to focus on the development of the best possible sensory and nutritional qualities of food from sustainable plant proteins to feed the ever-increasing global population.

Local nuts and legumes are important staple foods and are key component of Mediterranean cuisine and culture. Modern technology and the demand for healthier, efficient, and sustainable food systems can bring the use of local nuts and legumes to a new era by using traditional knowledge and modern technology to enrich food, diversify agriculture and generally create the sustainable intensification of Mediterranean food systems.

## Figures and Tables

**Table 1 foods-11-03858-t001:** Ready-to-eat products formulated with legumes and nuts available in the bibliography.

Food Type	Raw Material	Formulation	Elaboration	Final Product	Target Audience	Country	Observations	Reference
RTE baby food	Legumes (*cowpea and peanut*)	Cowpea, ripe banana, and peanut (47%, 40% and 13%, respectively)	Cowpea, ripe banana, and peanut were weighed, mixed, and milled (E Grinding mill). Final product was passed through the mill until the desired particle size was obtained. The product was stored in sealed containers at −18 °C	Weaning food	Babies	Raw material from Georgia, U. S	Positive results have been obtained to produce foods for weaned infants that satisfy their nutritional needs with 16.89% protein and 8.38% oil	[68]
RTE breakfast products	Legumes (*Soybean seeds*)	Mixed flour (600 g of maize and 400 g of soy flour), 100 g of cassava starch, 225 g of sugar and 12 g of salt. Pineapple, pawpaw, and banana ripe fruit pulps were added separately to the breakfast formulation at concentrations of 0, 100, 200, 300 and 400 g/kg flour.	Soybean seeds and maize grains were converted into flour, cassava roots into starch and fruits into pulp. The ingredients were completely mixed, steamed for 1 h, sliced into thin slices, dried at 55 °C, milled into powder, cooled, packaged, and stored in the freezer during the period of chemical and sensory analysis. To reconstitute the breakfast cereal, 200 mL water at 95 °C was mixed with 100 g each product (ratio of 2:1)	A RTE breakfast cereal with fruit pulp	Young audience and adults for the “on-the-go” consumption	Nigeria	After tasting, the sample of 100 g of fruit pulp per kg of flour (7% of total weight) was the most acceptable among tasters.	[69]
RTE breakfast	Legumes (*Popped pearl millet*)	Popped pearl millet (29.2–42%), popped amaranth (12.0%), puffed wheat (3.2–10%), flax seeds (0.0–7.0%), sunflower seeds, raisins (4.0%), honey (2.2%), sugar (20%), sunflower oil (6.3%) and water (6.3%).	Dry ingredients were mixed in a stainless-steel bowl. Sugar syrup was prepared using sugar and water (50–60 °C) with addition of sunflower oil and honey. Then, syrup mixture was added to the dry ingredients and mixed until it was homogenously dispersed. Final product was baked (50–60 °C for 15 min). The breakfast cereal was cooled at room temperature (RT:27 °C), packaged and stored at RT	Healthy ready-to-eat breakfast cereal from popped pearl millet	All audiences looking for healthier options	India	The pearl millet RTE-breakfast cereal contained high number of proteins, dietary fibre, folic acid and minerals such as calcium, phosphorus and iron	[70]
RTE breakfast	Nut (*Bambara groundnut*)	Bambara groundnut, malted sorghum, pearl millet and banana were processed into flours and mixed in the ratio of 50:20:20:10, respectively.	Ingredients were mixed and extruded, using the response surface methodology, screw speed at 300–350 rpm, barrel temperature at 180–220 °C, and feed moisture at 12–16%	Bambara groundnut (*Vigna subterranea*)-based ready-to- eat breakfast cereal	All audiences looking for healthier options	Nigeria	It was found that the Bambara variety of groundnut flour can be used to produce a RTE breakfast cereal using extrusion. It contains 17.24% protein and better digestibility. After a sensory analysis, it has been well accepted and in addition the microbiological analysis has been positive.	[71]
RTE drink	Legumes (*mung* *bean starch and* *soybean flour*)	Drink with Mung bean starch: Mung bean starch (45.37%), soybean flour (39.96%), groundblack sesame seed (4.01%), acesulfame-K (0.1%), rice bran oil (1.05%), and sugar (9.51%) Drink with polished rice + brown rice flour: rice flour (26.67%), brown rice flour (26.67%), soybean flour (28.54%), ground black sesame seed (4.07%), acesulfame-K (0.1%), rice bran oil (2.44%), sugar (9.51%)	Dried ingredients were mixed with water (40 g/100 g, respectively). Then, the slurry was applied onto a pilot-scale double-roller drum dryer (30 cm diameter and 45 cm length) with 0.1 mm roller gap, 1 rpm roller speed and 6–9 kg/cm^2^ steam pressure. The drum-dried sheets were broken into 1 cm size and dry-mixing with soybean flour, sugar, and acesulfame K. The dehydrated products were packed in heat-sealed polypropylene (PP) bags until use	Cereal and legume based instant drink	Elderly	Thailand	The products (a_w_ < 0.3) had balanced energy distribution, good quality protein, and energy from saturated fat < 8 kcal/100 kcal and free sugar < 10 kcal/100 kcal	[72]
RTE instant soup	Legumes (*mung bean starch and soybean flour*)	Soup with Mung bean starch: Mung bean (41.1%) starch, soybean flour (32.06%), ground black sesame seed (4.11%), rice bran oil (1.23%), soup seasoning (21.5%) Soup with polished rice + brown rice flour: rice flour (24.79%), brown rice flour (24.79%), soybean flour (22.31%), ground black sesame seed (4.13%), rice bran oil (2.48%), soup seasoning (21.5%)	Dried ingredients were mixed with water (40 g/100 g, respectively). Then, the slurry was applied onto a pilot-scale double-roller drum dryer (30 cm diameter and 45 cm length) with 0.1 mm roller gap, 1 rpm roller speed and 6–9 kg/cm^2^ steam pressure. The drum-dried sheets of different formulas were broken into 1 cm size and mixed with shitake-flavoured powder. The dehydrated products were packed in heat-sealed PP bags until use	Cereal and legume based instant soup	Elderly	Thailand	The products (a_w_ < 0.3) had balanced energy distribution, good quality protein, and energy from saturated fat < 8 kcal/100 kcal and free sugar < 10 kcal/100 kcal	[72]
RTE plant-based cheese	Legume (*Soy*) Nut (*Cashew*)	Vegan milk cheese was prepared by substituting soy milk with cashew nut milk at different proportions: 0%, 20%, 40%, 60%, 80%, and 100%.	First, the kernel of the cashew nut was cracked to remove the nut. Nuts were cleaned and soaked overnight. The dry soybeans were soaked in clean water for 20 min and then boiled for 30 min. The beans were cooled and then dehulled. Nuts and beans were wet milled, after which the slurry was sieved using a muslin cloth to obtain the milk. Soy milk was substituted for cashew milk at different proportions. The milk was boiled for 5 min with occasional stirring followed by cooling to 78 °C. The coagulant solution made by dissolving 15.0 g alum in 25 mL of distilled water was added. The homogenate was kept for 30 min for effective coagulation. The curd was drained in a muslin cloth and pressed for 45 min using a weight of 6 kg. Cheese was weighed and cut into rectangular shapes. It was then boiled with salt, pepper, and seasoning cube (all to taste)	Cheese analog	All audiences, lactose intolerant	Nigeria	The overall acceptability of the samples decreased with increasing cashew nut milk substitution	[73]
RTE plant-based milk	Legumes (*Lentil*) Nut (*Almond*)	Plant-based milk substitute (Almond) with lentil protein isolate in sunflower oil (3.3%), sucrose (2.5%) and salt (0.08%)	Milk was elaborated with lentil protein isolate (3.3%) in sunflower oil (treated at 900 bar and 85 °C), and commercial almond-milk. Sucrose (2.5% *w*/*w*), salt (0.08% *w*/*w*) were added to the formulation	Lentil-based milk	All audiences, lactose intolerant	Ireland and Germany	After performing sensory analyses of the lentil-based milk replacer, it has comparable organoleptic profiles to other commercial plant-based milks.	[74]
RTE porridge	Legumes (*fava bean*)	Prosso millet, little millet, soaked fava beans and germinated fava beans in different proportions: P1:70-0-30-0; P2: 0-70-30-0; P3: 70-0-0-30); P4: 0-70-0-30; P5: 35-35-30-0; P6: 35-35-0-30, respectively	All the grains were washed with tap water. The grains were soaked in the water (1:3) at 65 °C for 3.5 h followed by steaming for 15 min at 121 °C and 15 psi. The cooked grains were placed in the hot air oven (45 °C) till desired moisture content (8%), followed by milling and passed through 70 mesh sieve size. For the formulation of RTE porridge the millets (little and prosso) and fava beans (soaked and germinated) were mixed in different proportions. For germination, fava beans were soaked (200 g/600 mL) in water for 24 h and kept further for 24 h at 25 °C in a dark place as a single grain bed	Ready to mix gluten-free porridge	All audiences, including gluten intolerant people	India	The in vitro starch digestibility of formulation P5 (little millet, prosso millet and soaked fava bean) was significantly higher (67.70%) compared to other formulations.	[75]
RTE purée	Legumes (*Kabuli and Apulian chickpea*)	100% chickpea	Chickpeas were soaked overnight in tap water (1:2 *w*/*v*) at room temperature (23 °C). Then, soaking water was discarded, and softened chickpeas were boiled in tap water (1:2 *w*/*v*) for 70 min. After cooking, the chickpeas were ground by a blender, and the obtained purée was poured into glass jars (125 mL). The sterilization process (F0 ¼ 3) of the jars was performed in an autoclave.	Canned chickpea purée	All audiences	Italy	Looking at all these features make canned puree of chickpeas a healthy RTE food, which is at the same time rich in fiber and bioactive compounds, and able to fulfil certain needs generated by the timesaving needs of modern lifestyle.	[76]
RTE purée/cream	Legumes (*Chickpea flour*)	Chickpea flour, water, extra virgin olive oil, soymilk, common salt and lemon juice	For slurries, chickpea seeds were hydrated with tap water (1:6 *w*/*v*) and boiled (under 98.07 kPa for 20 min). The cooking water was drained used for chickpea slurries. Seeds were discarded. Chickpea flour toasting was performed at 90 °C for 20 min using a TM 31 food processor. Four chickpea flour (CF) slurry formulations were prepared: with raw (RCF) or toasted flour (TCF), and with lemon juice (RCFL, TCFL) CF slurries without added lemon juice were prepared from 14.29% of raw or toasted flour, 57.14% of cooking water, 27.43% of soymilk, 0.57% of oil, and 0.57% of salt. In the CF slurries with added lemon juice, the total amount of lemon (0.56%) was subtracted from the original soymilk content. Formulations were unpressurized and pressurized at 600 MPa and 50 °C for 15 or 25 min (HPP-induced CF products). Homogenates were irradiated for 2 min at an output power rating of 700 W.	RTE chickpea flour purée or cream	All audiences looking for healthier options	Spain	All the CF products were microbiologically innocuous and stable during two months at cooled storage. Mainly, the HHP-treated chickpea flour (CF) products diverged in their texture depending on the CF used, the holding time and the presence of lemon juice, whereby each individual product could be categorised as a CF purée or a cream. Moreover, all the formulations exposed similar very high sensory quality.	[77]
RTE snack	Legumes (*Pulse flour (red lentil)*)	Variable ingredients: Rice flour (60–80%) and pulse flour (10–30%); and fixed carrot pomace (10%) and salt (2%)	First, pomace carrot was treated with 1% (*w*/*v*) citric acid at 65 °C, and dried to moisture content of 6.0% (db). Pomace flour was obtained by grinding dried pomace in a 750 W grinder. Rice and pulse samples were ground to elaborate flour. Variable and fixed ingredients of the formulation were mixed and extruded at 40, 70 and 100 °C. Barrel diameter and length/diameter ratio were 2.5 mm and 16:1, respectively.	Cereal-based RTE expanded product formulated with carrot pomace	Young audience and adults for the “on-the-go” consumption	India	The ideal extrusion method factors obtained were an 80:10:10 (rice flour/pulse flour/carrot pomace powder) sample formulation.	[78]
RTE snack	Legumes (*Pearl millet,* *Green gram, Soya bean*)	The ready-to-eat nutritious snack mix was developed by blending the flour from popped millets and legumes with sugar and other ingredients in the optimized proportion 30:20:27:23. Sorghum, grain amaranthus, bengal gram, Pearl millet, Green gram, Soya bean, sugar, honey, sunflower oil, skimmed milk powder and groundnut seeds	Sorghum and pearl millet (5 Kg of each) were dried to a moisture content of 18%. The tempered grains were popped by high temperature and short time (HTST) treatment (230 ± 5 °C). Amaranthus was popped by subjecting the grains to direct heat in a pan. For popping the legume components, about 5 kg of each of split green gram and soya were steeped in water at ambient temperature for 2 h, followed by steaming at atmospheric pressure for 20 min. The steamed grains were dried in a mechanical drier HTST. Groundnut seeds were toasted at 70 °C. Popped sorghum, pearl millet, grain amaranthus, green gram, soya and bengal gram were pulverized to flour of (355 μm). The toasted groundnut seeds were disintegrated to smaller grits (1204 μm). Accordingly, the RTE snack mix was prepared by blending 10 g popped sorghum, 10 g popped bajra, 10 g popped amaranthus, 8 g popped green gram dhal, 8 g popped bengal gram dhal, 4 g popped soy dhal, 6 g skimmed milk powder, 3 g roasted groundnut seeds, 27 g sugar, 9 g oil, 2 g honey, 1 g each of pectin, edible gum, and glycerol in a ribbon mixer/blender.	A RTE nutritious snack mix.	Young audience and adults for the “on-the-go” consumption. The product can be mixed with desired quantity of water, or with milk to prepare porridge.	India	The sensory evaluation of the product revealed that colour, taste, texture, aroma, appearance, and overall quality were in satisfactory range with mean score of 6.8.	[79]
RTE snack	Legumes (*laird lentil, yellow split chickpea and peanut*)	Cereal-legume composite bars contained the same fixed ingredients (oats, laird lentils, yellow split chickpeas, dates, honey, water), one of two variable cereals (maize or semolina) and one of three add-on ingredients (toasted peanuts and dates, toasted sesame seeds and dates, or oat flakes)	Maize (30 g) or semolina (30 g), oats (30 g), laird lentils (40 g) and yellow split chickpeas (40 g) were individually toasted in the oven (Bosch) at 130 °C for 60 min. The toasted ingredients were then ground in a mixer (Robot Coupe) followed by the addition of cardamom powder (1 g). Ten dates were finely chopped and incorporated into the mixture with 50 mL of warm water and 35 mL of honey. The mixture was then uniformly spread in a rectangular pan (20 cm × 8 cm) to a thickness of 2 cm and sprinkled with either toasted peanuts and dates, toasted sesame seeds and dates, or oats flakes, thus yielding six different cereal-legume formulations. The bars were cooled for 30 min and cut into bars (8.5 cm × 2.5 cm).	Novel pulse-based snack bar	Young audience and adults for “on-the-go” consumption	Mauritius	The storage study indicated a reasonably short shelf-life of <2 days when kept at 4 °C.	[80]
RTE snack	Legumes (*laird lentil, yellow split chickpea and peanut*)	Cereal-legume-vegetable composite bars contained the same fixed ingredients (oats, laird lentils, yellow split chickpeas, dates, honey, water), one of two variable vegetables (beetroot or carrot) and one of three add-on ingredients (toasted peanuts and dates, toasted sesame seeds and dates, or oat flakes).	A beetroot or carrot was washed, peeled, and grated. Grated beetroot or carrot (30 g), oats (30 g), laird lentils (40 g) and yellow split chickpeas (40 g) were individually toasted in the oven at 130 °C for 30–60 min depending on the ingredient. The toasted ingredients were then ground in a mixer followed by the addition of cardamom powder (1 g). Dates were chopped and incorporated into the mix with 50 mL of warm water and 35 mL of honey. The paste was spread in a rectangular pan (20 cm × 8 cm) to a thickness of 2 cm and sprinkled with toasted peanuts and dates or toasted sesame seeds and dates or oat flakes) resulting in six different formulations. The bars were then cooled for 30 min and subsequently cut into bars (8.5 cm × 2.5 cm).	Novel pulse-based snack bar	Young audience and adults for the “on-the-go” consumption	Mauritius	The storage study indicated a relatively short shelf-life of <2 days when kept at 4 °C.	[80]
RTE snack	Legumes (*Black gram*)	Extruded RTE snacks were prepared from flour blends made with corn flour, Bengal gram flour, roots and tuber flours in a proportion of 60–80:20:20 respectively and moisture was adjusted to 17–20%.	Tubers, corn and black gram were peeled, washed and cut into 1–2 cm cubes. Tubers were soaked in sodium metabisulfite (0.075%) and oven dried at 50 °C for 30 h. Corn and black were dried at 5 °C for 20 h. Then, all ingredients were milled into flour and sifted a 300 µm sieve. Ingredients were mixed and extruded in a co-rotating twin screw extruder. The barrel diameter and L/D ratio were 37 mm and 27:1, respectively. Different formulations were extruded at 80 ± 5 °C and 95–105 °C temperature, 300–350 rpm screw speed, 100 ± 10 °C die temperature and 15 ± 2 kg/h feed rate.	Extruded RTE snack	All audiences looking for healthier options	India	The fibre and energy content of the RTE extruded snack enhanced in experimental samples prepared using root and tuber flours.	[81]
RTE snack	Legumes (*mung bean starch and soybean flour*)	Snack with Mung bean starch: Mung bean starch (51.55%), soybean flour (38.32%), ground black sesame seed (4.56%), acesulfame-K (0.1%), rice bran oil (0.91%), sugar (4.56%) Snack with polished rice + brown rice flour: rice flour (31.16%), brown rice flour (31.16%), soybean flour (25.67%), ground black sesame seed (4.58%), acesulfame-K (0.1%), rice bran oil (2.75%), sugar (4.58%)	Dried ingredients were mixed with water (40 g/100 g). The slurry was applied onto a pilot-scale double-roller drum dryer, 30 cm diameter, 45 cm length, a hard chrome material on the roller surface, with 0.1 mm roller gap, 1 rpm roller speed and 6–9 kg/cm^2^ steam pressure. The drum-dried sheets of different formulas were broken into approximately 1 cm size and used directly for the flake snack. The products were packed in heat-sealed PP bags.	Cereal and legume-based snack	Elderly	Thailand	The products (a_w_ < 0.3) had well-adjusted energy distribution, good quality protein, and energy from saturated fat < 8 kcal/100 kcal and free sugar < 10 kcal/100 kcal.	[72]
RTE snack	Legumes (*Common bean*)	Flour portion (%) of 1:1:2.5 (wheat: maize: common bean), 1.7% eggs, 2.5 white sugar, 1.5% butter, 0.1% salt and 0.1% yeasts.	All the ingredients were weighed and added in a bowl. After mixing the ingredients, common bean flour, sugar, salt and yeast, the eggs, and the melted butter were added and converted to dough. Finally, the dough was cookie-shaped and baked at 180 °C for 10–15 min.	What cookies	All audiences looking for healthier options	Portugal	This study revealed, for the first time, through a human intervention trial the importance of using legumes (common beans in specific) as alternative ingredients to increase ready-to- eat products’ nutritional quality.	[82]
RTE snack	Legumes (*chickpea flour and peanut*) Nuts (*walnuts*)	Black wheat (30%), corn flour (20%), flour (15%), chickpeas (15%), black beans (2%), peanuts (3%), melon seeds (3%), walnut (3%), black sesame seeds (4%), red date powder (15%), sugar powder (15%), maltodextrin (15%), and sunflower oil (20%)	Ingredients were pre-treated and mixed. Edible additives (maltodextrin, suspension beverages stabilizer, other nutritional supplements) were added to de mixture. The homogenate was dried and packaged.	Healthy nutritional product	All audiences looking for healthier options	China	BWGP has several benefits, which include: a regional distinguishing, simple creation methods, and rich flavours. Overall, our study intensive on the nutritional, flavour, and compositional effects and how a food product can be prepared healthier, more sustainable or more acceptable to the consumer.	[83]
RTE snack	Legumes (*Common beans*)	White maize (70%), common bean (30%)	White maize or common bean kernels (1 kg) were grinded to grits (0.425 mm). Grits were conditioned with purified water until 18% moisture; each lot was packed in a polyethylene (PE) bag and stored at 4 °C for 12 h. Extrusion cooking was carried out in a lab-scale extruder (Model 20 DN). Expanded snacks were produced at 164 °C and screw speed of 187 rpm. Final products were cooled, equilibrated (25 °C, RH = 65%, 1 h) and packed in hermetic plastic bags.	RTE extruded snack	All audiences looking for healthier options	Mexico	The expanded snack could be source of bioactive, nutritional and antioxidant compounds for the upgrading of the consumer’s health.	[84]
RTE snack	Nuts (*Baru almonds (BA), Brazil nuts (BN)*)	BA 100, BN25:BA75, BN50:BA50, BN75:BA25, and sunflower lecithin 2%, Brazil nuts oil 3%, Honey 9%	BN and BA were mixed in a food processor for 15 min, followed by the addition of Brazil nuts oil, lecithin, and honey. The obtained mass was laminated and pressed at a pressure of 4.32 Pa, packed with a plastic film, and refrigerated for 8 h.	Nutritive bar	Young audience and adults for the “on-the-go” consumption	Brazil	The highest satisfactoriness indexes for taste and texture (59 and 67%, respectively) were detected for BN25:BA75 while BA100 achieved the maximum acceptability index for odour, colour, and global perception (71, 73, and 72%, respectively).	[85]
RTE supplemented food	Legumes (*Green gram*)	Germinated wheat, green gram, potato flour, spinach leaves powder, baking powder and baking soda (52.5:15:30:2.5) were used. Fat was added and creamed. Sugar and milk were also used	Wheat and green gram were germinated, dried, and converted into flour. Potatoes were washed, peeled, sliced, boiled, dipped in potassium metabisulphite solution, dried, and grounded into flour. Spinach leaves were washed, dried, and made into powder. Ingredients were mixed and sieved into the creamed fat. Powdered sugar (52 g) was added to the creamed mixture. Smooth dough was prepared using milk (10 mL) with dissolved ammonium bicarbonate. The smooth dough was rolled to 1/4-inch thickness. The required shapes were cut out of the dough with a cutter and baked at 150 °C for 20 min.	Biscuits	Malnourished children	India	Biscuits were adequate at 30 per cent level of potato flour and 2.5 per cent level of spinach leaves powder.	[86]
RTE snack	Legumes (*carob fruit and pea*)	Pea, rice, carob, salt, calcium carbonate. Different formulations: 1rst (20:80:0:0.5:0.5), 2nd (20:75:5:0.75:0.75), 3rd (20:70:10:1:1), 4th (40:60:0:1:0.5), 5st (40:55:5:1:0.75), 6st (40:50:10:1:1)	The seeds were milled and passed through a 1-mm sieve. Blends were prepared by mixing the different ingredients. The flours were blended in a domestic mixing system and stored in PP bags until needed. The extrusion was developed at 125 °C and 900–950 rpm.	Snack	Gluten-free expanded product	Spain	The results achieved verified that combinations based on legumes (carob fruit and pea) and rice can be a fresh source of bioactive compounds to be used in the elaboration of expanded gluten-free snack.	[87]
RTE snack	Legumes (*carob fruit and bean)*	Rice (70–55%), bean (20 or 40%) and carob fruit (5 or 10%)	Flour blends were prepared by mixing rice (70–55%), bean (20 or 40%) and carob fruit (5 or 10%) flours to obtain 6 formulations (20.0, 20.5, 20.10, 40.0, 40.5 and 40.10). The first number of the code sample corresponded to bean percentage (20 or 40) and the second one (0, 5 or 10) to whole carob fruit percentage present in the formulation. Ingredients were mixed and extruded at 125 °C and 900–950 rpm.	Snack	Gluten-free expanded product	Spain	After extrusion, a decrease in the total dietary fibre (20–25%), and a reorganization of the fibre portions was observed.	[88]

**Table 2 foods-11-03858-t002:** Technologies used in the literature to modify legume proteins for the improvement of technological properties.

Ingredient	Technological Property	Modification	Result	Possible Application Examples	Reference
Lentil flour	Water and fat binding, emulsifying, foaming, gelling, and texturizing	Heat treatment (dispersed in Millipore water under agitation for 1 h at 20 °C, boiled in a water bath at 90 °C for 20 min)	Reduction in solubility (due to aggregation)	Meat analogues: nuggets, sausages, meat balls; cake doughnut; cookies; vegan cheese; vegan ice cream	[95]
Soybean protein isolate	Emulsifying, fat and water absorption, thickening, gelling, foaming, and film formation	Heat treatment (95 °C for 15 and 30 min)	Improved emulsifying capacity	Bakery products: cakes, pancakes, bread, doughnuts; gravies and soups; Pizza; Meat analogues; vegan cheese, vegan ice cream	[49]
		High pressure treatment (600 MPa)	Reduced solubility		[96]
		Extrusion-cooking (150 rpm at 137–160 °C)	Reduced solubility		[97]
		Glycosylation (polysaccharides)	Improved thermal stability, viscosity, solubility, emulsifying, foaming, and water holding capacity		[98]
Soybean flour	Emulsifying, fat and water absorption, thickening	Heat treatment (dispersed in Millipore water under agitation for 1 h at 20 °C, boiled in a water bath at 90 °C for 20 min)	Reduction in solubility (due to aggregation)	Meat analogues: Frankfurters, sausages, meat patties; bakery products; Soups	[95]
Soybean β-conglycin and glycin	Emulsifying, fat and water absorption, thickening, gelling, foaming, and film formation	Ultrasounds (20 kHz at 400 W for 5, 20 and 40 min)	Increased solubility, emulsifying capacity, emulsion stability and surface hydrophobicity	Bakery products; Egg-free products; Mayonnaise	[99]
Pea protein isolate	Emulsifying, fat and water absorption, thickening, gelling, foaming, and film formation	Heat treatment (95 °C for 15 and 30 min)	Improved emulsifying capacity	Bakery products; Egg-free products; Mayonnaise	[49]
		Acetylation (succinic anhydride n-octenyl succinic anhydride and dodecyl succinic anhydride)	Improved solubility		[100]
		Enzymatic cross-linking	Increased gel strength		[101]
Pea legumin	Emulsifying, fat and water absorption, thickening, gelling, foaming, and film formation	Heat treatment followed by fast cooling	Improved gelling capacity	Meat analogues; Bakery products; Egg-free products; Mayonnaise	[102]
Black bean protein isolate	Emulsifying, fat and water absorption, thickening, gelling, foaming, and film formation	Ultrasounds (12 and 24 min at 150, 300 and 450 W	Increased solubility	Bakery products; Egg-free products; Mayonnaise	[103]
Pea protein concentrate	Emulsifying, fat and water absorption, thickening, gelling, foaming, and film formation	High pressure treatment (350 and 550 MPa)	Improved gel strength	Meat analogues; Bakery products; Egg-free products; Mayonnaise	[104]
		Glycosylation (Arabic gum)	Improved solubility and emulsifying capacity	Bakery products; Egg-free products; Mayonnaise	[105]
Peanut protein isolate	Emulsifying, fat and water absorption, thickening, gelling, foaming, and film formation	Phosphorylation (55 °C and 5 h)	Increased solubility	Bakery products; Egg-free products; Mayonnaise	[106]
Mung bean protein isolate	Emulsifying, fat and water absorption, thickening, gelling, foaming, and film formation	Acetylation (succinic anhydride)	Improved solubility and formability	Bakery products; Egg-free products; Mayonnaise	[107]
Chickpea protein isolate	Emulsifying, fat and water absorption, thickening, gelling, foaming, and film formation	Alcalase enzyme (5% *w*/*v*) substrate, pH 8.0, 50 °C, 24 hr, Degree of hydrolysis 1% to 10%	Increased solubility and foaming capacity with degree of hydrolysis. Emulsion stability and foam stability decreases with degree of hydrolysis. Emulsion activity index improves with degree of analysis	Bakery products; Egg-free products; Mayonnaise	[108]
Chickpea flour	Water and fat binding, emulsifying, foaming, gelling, and texturizing	Heat treatment (dispersed in Millipore water under agitation for 1 h at 20 °C, boiled in a water bath at 90 °C for 20 min)	Reduction in solubility (due to aggregation)	Meat analogues: Nuggets, sausages, meat balls; cake doughnut; cookies; vegan cheese; vegan ice cream	[95]
Faba bean protein	Emulsifying, fat and water absorption, thickening, gelling, foaming, and film formation	Hydrolysis with different proteases (2–16% DH)	Improved solubility, foaming capacity, oil holding capacity	Meat analogues; Bakery products; Egg-free products; Mayonnaise	[109]
Common beans (White Aura & Red Toska)	Gelling, pasting properties.	Extrusion-cooking (twin-screw extruder with a screw length/diameter ratio of 24:1. Temperatures 50–130 °C, screws speeds: 300–700 rpm).	Improved viscosity, higher water-holding capacity, gel formation decreases.	Meat analogues; snacks or instant food components and additives	[110]
Faba beans proteins	Solubility, foaming, and emulsifying.	Acetylation of Faba Beans Protein Isolates (degree ~ 97%)	Improved solubility, higher protein levels, improved emulsifying capacity.	Plant-based emulsifiers.	[111]
Lupins	Gelling properties, Solubility	Ultrasound: Low frequency (16–100 kHz) high-intensity waves (10–1000 W/cm^2^)	Improved gelling properties, higher water holding capacity.	Light dairy products, desserts, and meat analogues.	[53]
Pea Protein Isolate	Gelling properties, emulsifying capacity	Ultrasound: Low frequency (20–100 kHz) high-intensity waves (10–1000 W/cm^2^)	Enhanced emulsifying capacity, nanoemulsion formation.	Food industry plant-based emulsifiers.	[112]

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
