# Peer review of "Valorization of Local Legumes and Nuts as Key Components of the Mediterranean Diet"

_foods, 2022, doi:10.3390/foods11233858_

Round 1

Reviewer 1 Report

This is a regular review concerning legumes. First thing is that authors do not focus too much on pro-healthy aspects of the kind of food they review, so the title should be changed. Second thing is that the manuscript should be adjusted to journal requirements. Besides, text has serious faults. Details below.

Line 86-89

You give some examples of Mediterranean legumins in Spain, what about Portugal, Italy, Greece, Israel examples?

137-138

Something wrong with this statistics. Original quotation is: Nearly 80% of proteins and 90% of calories consumed by humans in developing countries are supplied by plant products.

Line 144 vs 160 vs 172

There is an inconsistency as for world production legumins, 5 or 4 million tons, and in EU 4.11 million tons. Another wrong data concerns cereals (line 161) - worldwide cereal production 8 million tons? Think critically: this equals 8 billion kg, which means 1 kg per capita per year! 8 million tons is true, but for Czechia. E.g. Canada production of cereals is approx. 65 mln tons and Poland 35 mln tons.

Latin names should be italic – line 239 as an example

Line 415

What the question mark is doing here?

Table 2

Replace B in B-conglycin by Greek latter beta

Author Response

Dear reviewer,

The following revisions have been considered for improving the paper. 

Please, find below answers and corrections addressed.

Reviewer 1: This is a regular review concerning legumes. First thing is that authors do not focus too much on pro-healthy aspects of the kind of food they review, so the title should be changed. Second thing is that the manuscript should be adjusted to journal requirements. Besides, text has serious faults. Details below.

Answer: The authors are agreeing with the title change, so it has been done. The journal requirements have also been adjusted in consequence.

Line 86-89

You give some examples of Mediterranean legumins in Spain, what about Portugal, Italy, Greece, Israel examples?

Answer: In agreement with this, more examples have been selected and added with and without a specific quality mark (Line 88).

137-138

Something wrong with these statistics. Original quotation is: Nearly 80% of proteins and 90% of calories consumed by humans in developing countries are supplied by plant products.

Answer: The sentence has been modified in order to show that legumes provide an important amount of consumed protein percentage from plant-based diets. (Line 185).

Line 144 vs 160 vs 172

There is an inconsistency as for world production legumins, 5 or 4 million tons, and in EU 4.11 million tons. Another wrong data concerns cereals (line 161) - worldwide cereal production 8 million tons? Think critically: this equals 8 billion kg, which means 1 kg per capita per year! 8 million tons is true, but for Czechia. E.g. Canada production of cereals is approx. 65 mln tons and Poland 35 mln tons.

Answer: Quantities have been modified, and percentages of both legumes and cereal production lands are now presented (Lines 192 and 208).

Latin names should be italic – line 239 as an example

Answer: All the latin names have been changed.

Line 415

What the question mark is doing here?

Answer: Typing error deleted.

Table 2

Replace B in B-conglycin by Greek latter beta

Answer: Done, thank you for the comment

Reviewer 2 Report

All necessary corrections, questions and comments are indicated in the document.

The report is general in gathering information about legumes and nuts. 1) The work does not include a methodology, which is very important so that the reader can understand the period of time for which the review is made, how the information was searched for, what keywords were used, etc. In the review cannot use such a broad time frame, they use literary sources from 1962 and compares them with today. 2) It is also necessary to look at this application against the consumer, the countries and other criteria where these legumes grow, as this will also have a great impact on the composition and other parameters/technological processes. 3) There are also a lot of errors in the review, which point to low-quality work organization. 4) If they talk about the importance of legumes in the diet, then there must also be a summary of their composition and evaluation.

Author Response

Dear reviewer,

All the answers and comments have been addressed to improve the manuscript.

Please find below the answers.

Reviewer 2: All necessary corrections, questions and comments are indicated in the document.

Answer: Corrections and comments indicated in the document have been attended as it can be shown in the submitted new version.

1) The work does not include a methodology, which is very important so that the reader can understand the period of time for which the review is made, how the information was searched for, what keywords were used, etc. In the review cannot use such a broad time frame, they use literary sources from 1962 and compares them with today.

Answer: A methodology section has been added, indicating the literature review process, used keywords and inclusion and exclusion criteria for RTE food formulations and production processes literature.

2) It is also necessary to look at this application against the consumer, the countries, and other criteria where these legumes grow, as this will also have a great impact on the composition and other parameters/technological processes.

Answer: A mayor literature review has been assessed in order to add information about adequate biodigestibility information, also how to deal with the present anti-nutrients in legumes and nuts, as diverse processes can be assessed to reduce or eliminate these compounds. In addition, a further conclusion about the importance of evaluate the nutrimental intake and its bioavailability for consumers was aggregated.

3) There are also a lot of errors in the review, which point to low-quality work organization.

Answer: Corrections and comments indicated in the document have been attended to provide a better comprehension of the information and typing errors were suppressed.

4) If they talk about the importance of legumes in the diet, then there must also be a summary of their composition and evaluation.

Answer: More information about nutrimental content and anti-nutrients was added. This to fulfill the comment on their composition.

Round 2

Reviewer 1 Report

The paper has been corrected according to my remarks. I accept all explanations